# PICK-AND-UNFOLD: MODE-AWARE NON-LINEAR TUCKER AUTOENCODER FOR TENSORIAL DEEP LEARNING

## ABSTRACT

High-dimensional data, particularly in the form of high-order tensors, presents a major challenge in deep learning. While various deep autoencoders (DAEs) are employed as basic feature extraction modules, most of them depend on flattening operations that exacerbate the curse of dimensionality, leading to excessively large model sizes, high computational overhead, and challenging optimization for deep structural feature capture. Although existing tensor networks alleviate computational burdens through tensor decomposition techniques, most exhibit limited capability in learning non-linear relationships. To overcome these limitations, we introduce the Mode-Aware Non-linear Tucker Autoencoder (MA-NTAE). MA-NTAE generalized classical Tucker decomposition to a non-linear framework and employs a Pick-and-Unfold strategy, facilitating flexible per-mode encoding of high-order tensors via recursive unfold-encode-fold operations, effectively integrating tensor structural priors. Notably, MA-NTAE exhibits linear growth in computational complexity with tensor order and proportional growth with mode dimensions. Extensive experiments demonstrate MA-NTAE's performance advantages over DAE variants and current tensor networks in dimensionality reduction and recovery, which become increasingly pronounced for higher-order, higher-dimensional tensors.

## 1 INTRODUCTION

High-order tensors (multi-way arrays indexed by multiple coordinates) serve as the fundamental representation for modern data-intensive applications across scientific and industrial domains (Fu et al., 2022). Multi-view images (Lou et al., 2025), hyperspectral data (Xu et al., 2019), and spatio-temporal signals (Gong et al., 2023) *etc.*, all naturally manifest as tensors. These data structures preserve multidimensional relationships through distinct mode axes capturing wavelength, spatial coordinates, temporal frames, viewpoints, or sensor modalities. The exponential growth of such data has intensified the demand for learning models capable of compressing, mining, and analyzing high-order tensors.

Modern deep autoencoders (DAE) based on Multi-layer perceptions (MLPs) (Hinton & Salakhutdinov, 2006), including variants like Variational AEs (Kingma & Welling, 2014) and Adversarial AEs (Makhzani et al., 2016), remain dominant in unsupervised representation learning (Hu et al., 2025; Lin et al., 2023). However, they suffer from two critical limitations when processing tensor-form data: i) **Mode-agnostic compression**: Flattening operations discard mode-specific statistical dependencies (e.g., temporal correlations versus spatial correlations), which leads to an optimization disaster in recovering structural information; ii) **Exponential parameter growth**: For an $N^{th}$-order tensor, a fully connected layer mapping flattened input to latent code requires parameters scaling with the multiplication of all input dimension sizes (See the third-order case in Figure 1.1a). This leads to a compromise in the input-data dimensionality among researches (Zhu et al., 2024; Wang et al., 2023), where models are also forced to reduce hidden and latent dimensionality to ensure stable convergence.

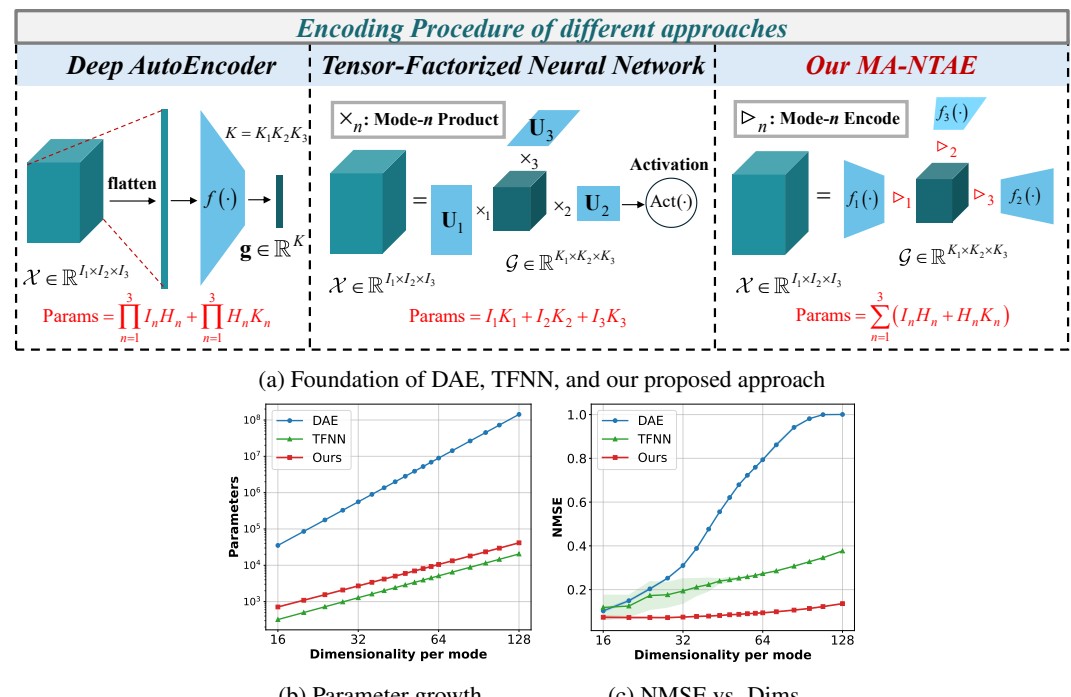

(a) Foundation of DAE, TFNN, and our proposed approach

(b) Parameter growth      (c) NMSE vs. Dims

Figure 1.1: Graphical abstract of our innovations and advantages over typical vector-based and tensor-based neural networks. Our MA-NTAE directly models the non-linear interactions between different modes, achieving better compression performance with less parameters. (b) and (c) are the results in third-order tensor scenarios (See Section 4.1 for details).

## 1.1 CLASSICAL TUCKER DECOMPOSITION REVISITED

A naive yet elegant remedy to overcome the curse of dimensionality is offered by the classical multi-linear algebra in **Tucker decomposition** Tucker (1966), which factorizes a tensor $\mathcal{X}$ into a core tensor $\mathcal{G}$ and factor matrices $\{\mathbf{U}^{(n)}\}_{n=1}^N$, achieving linear parameter growth in tensor order $N$ and proportional growth in mode dimensions. Through **unfold-encode-fold**, the structural information is naturally introduced and integrated into the low-rank approximation for tensor data. During the last decade, researchers have made an effort to utilize Tucker's principle and present tensor autoencoder networks (Liu & Ng, 2022; Chien & Bao, 2018; Luo et al., 2024). Among them, Chien & Bao (2018) successfully construct a common Tensor-factorized Neural Network (TFNN) to perform non-linear feature extraction (See Figure 1.1a). However, these approaches are inherently based on linear tensor decomposition frameworks, where neural networks primarily serve to learn the factor matrices for decomposing input data (raw inputs or feature tensors extracted by backbone networks). Although these methods introduce non-linear transformations by applying activation functions to the core tensor, they fail to effectively model the non-linear interactions between different modes, ultimately limiting the model's ability to learn complex cross-mode dependencies in the data.

## 1.2 OUR APPROACH: A NON-LINEAR TUCKER FRAMEWORK

Inspired by Tucker decomposition and existing tensor networks, we propose the **Mode-Aware Non-linear Tucker Autoencoder (MA-NTAE)**, an intuitive yet effective tensor neural network architecture. A foundation comparison of existing and our approaches is shown in Figure 1.1a. The overall framework of our approach is illustrated in Figure 3.1, which embodies three fundamental innovations: **1) Mode-aware non-linear encoding.** MA-NTAE extends Tucker decomposition through a recursively applied *Pick–Unfold–Encode–Fold* strategy. This approach effectively models interactions within individual modes while propagating learned representations across different modes to further explore cross-mode relationships. **2) Implicit structural priors.** Each time of mode-aware encoding exposes mode-wise covariance structures, where the encoder learns *non-linear Tucker fac-*

*tors* and the folded latent core becomes a dynamically optimized core tensor. By incorporating tensor-structured priors, the proposed method narrows the parameter optimization space, enabling faster and more stable deep mining of tensor data. **3) Low computational complexity.** MA-NTAE achieves scalable computational complexity that grows linearly with tensor order and proportionally with mode dimensions, while maintaining parameter efficiency.

Our main contributions are: 1) We propose a non-linear Tucker-driven framework that unifies classical tensor factorization with modern autoencoding and allows flexible mode-aware operations in tensorial deep learning. 2) We offer a simple yet effective principle—Pick-and-Unfold to handle the curse of dimensionality in higher-order tensor scenarios. 3) We provide extensive empirical evidence on synthetic and real tensors demonstrating superior tensor data representation in compression tasks, with advantages that amplify as data dimensionality grows.

## 2 RELATED WORK

**Notations**. Tensors are denoted by bold calligraphic letters ($\mathcal{X}$), matrices by bold capitals ($\mathbf{X}$), and vectors by bold lower-case letters ($\mathbf{x}$). $\mathbf{X}^{(n)} \in \mathbb{R}^{I_n \times \prod_{k \neq n}^{N} I_k}$ denotes the mode-$n$ unfolding of $\mathcal{X} \in \mathbb{R}^{I_1 \times \cdots \times I_N}$.

**Deep Autoencoders**. Deep Autoencoders (DAEs) have evolved significantly since their inception as linear dimensionality reducers (Bourlard & Kamp, 1988). Modern variants includes regularized AEs (Vincent et al., 2010; Rifai et al., 2011)and probabilistic AEs Kingma & Welling (2013); Makhzani et al. (2015). Despite these advances, all flatten high-order tensors into vectors, destroying multi-linear structure and inducing $\mathcal{O}(\prod_{n=1}^{N} I_n)$ parameter scaling. Although convolutional AE (Masci et al., 2011) greatly relieves computational burden, it is only applicable to tensors of up to third-order. Our work fundamentally differs by mode-wise recursive processing, which can be extended to tensors of any order.

**Tucker Decomposition**. Tensor decomposition extracts latent structures from high-order data through multi-linear algebraic (Kolda & Bader, 2009), which (Tucker, 1966) represents $\mathcal{X}$ as a core tensor $\mathcal{G} \in \mathbb{R}^{K_1 \times \cdots \times K_N}$ multiplied by factor matrices $\mathbf{U}_n \in \mathbb{R}^{I_n \times K_n}$ along each mode:

$$\mathcal{X} \approx \mathcal{G} \times_1 \mathbf{U}_1 \times_2 \cdots \times_N \mathbf{U}_N, \tag{1}$$

where $\mathcal{G} \times_n \mathbf{U}_n := \mathbf{U}_n \mathbf{G}^{(n)}$ is the mode-$n$ product. The multi-linear rank $(K_1, \ldots, K_N)$ in Tucker's allows mode-specific compression. Applications based on Tucker decomposition span multiple domains, including image compression (Ballester-Ripoll et al., 2020), signal processing (Haardt et al., 2008), and pattern recognition (Hua-Chun Tan & Yu-Jin Zhang, 2008). However, the linear algebra nature of Tucker decomposition inherently limits its broader application in modern complex downstream tasks.

**Tensorial Neural Network**. Recent advances in *tensorial neural networks* (TNNs) show that combining multi-linear algebra with deep learning produces compact, structure-aware models. Currently, TNNs can be roughly distinguished by their applicability as autoencoders. One branch of TNN research leverages tensor decomposition on features (extracted by other modules) that are deliberately structured into tensors using domain knowledge to obtain interpretable, domain-specific information. For example, Hyder & Asif (2023) combines tensor ring factorization with a deterministic autoencoder to impose low-rank structural constraints on the latent space, leveraging dataset articulations for improved compressive sensing tasks like denoising and inpainting. Zhao et al. (2024) tensorizes multi-view low-rank approximations so that inter-view and intra-class structures are learned jointly, boosting robust hand-print recognition. This type of TNNs, although preserving a valid foundation, pays less attention to the structural information of the original data. The other branch of TNNs serves as an autoencoder to extract features, which can directly preserve structural information from raw tensor data. For example, Novikov et al. (2015) utilizes tensor-train decomposition to compress MLP-based DAE and greatly reduces parameters. Chien & Bao (2018) replaces every MLP layer with a Tucker factorization followed by an activation function to form a non-linear approximation, preserving mode-wise correlations. Newman et al. (2024) develops a T-SVDM representation to efficiently parameterize MLP-based DAE while preserving third-order tensor structures.

Although TNN-based AEs make great progress in incorporating prior structural information for parameter reduction, both branches of TNNs' tensor decomposition processes remain fundamentally

Figure 3.1: Overall framework of our approach in third-order tensor scenarios. For a batch of tensor data (where each frontal slice represents one sample), we sequentially perform mode-$n$ Unfold–Encode–Fold procedure, progressively reducing dimensionality across modes. The decoding process follows the reverse mode order to reconstruct data matching the original input dimensions, after which we compute the reconstruction loss. To ensure training stability, skip connections from residual learning are incorporated between corresponding encoder-decoder pairs, enhancing the network's capacity for modeling high-order tensor data.

rooted in linear operations, incapable of achieving a fully non-linear decomposition of tensors that integrates deeper cross-mode dependencies. Building on this line, we propose a mode-aware tensor autoencoder that performs *Pick-Unfold–Encode–Fold* operations, realizing a flexible *non-linear Tucker compression* with enhanced ability to capture complex non-linear dependencies.

# 3 METHODOLOGY

In this section, we formalize the proposed *Mode-aware Non-linear Tucker Autoencoder* (MA-NTAE) and detail its optimization. Figure 3.1 provides an overview of our approach.

**Fundamental Problem**. The fundamental challenge we address involves developing an efficient tensor compression framework for high-order data representations. Given an $N$-th order tensor $\mathcal{X} \in \mathbb{R}^{I_1 \times I_2 \times \cdots \times I_N}$ ($N \geq 3$), our objective is to learn a non-linear mapping $\mathcal{X} \to \mathcal{G} \in \mathbb{R}^{K_1 \times \cdots \times K_N} (K_n < I_n)$ that preserves the intrinsic cross-mode structure while achieving dimensionality reduction. The traditional Tucker decomposition achieves multilinear mapping and reconstruction through a series of mode-specific linear encoders and decoders. Our proposed framework extends this concept to multi-non-linear scenarios by replacing the factor matrices with non-linear mappings:

$$\mathcal{G} = \mathcal{X} \triangleright_1 f_1 \triangleright_2 \cdots \triangleright_N f_N,$$
$$\hat{\mathcal{X}} = \mathcal{G} \triangleright_N g_N \triangleright_{N-1} \cdots \triangleright_1 g_1, \tag{2}$$

where $\mathcal{X} \triangleright_n f_n := \text{fold}_n \left( f_n \left( \text{unfold}_n(\mathcal{X}) \right) \right)$, and $f_n$ and $g_n$ are the mode-specific encoder and decoder sequences, respectively.

## 3.1 CORE ARCHITECTURE

**Pick–Unfold–Encode–Fold Recursion**. The compression mechanism employs a recursive Pick-Unfold-Encode-Fold procedure that selectively processes individual tensor modes. For an ordered set of target modes $\mathcal{S} = \{s_1, \ldots, s_L\} \subseteq \{1, \ldots, N\}$, each compression stage $\ell \in \{1, ..., L\}$ executes three key operations:

1) Mode-specific Unfolding: The current latent tensor $\mathcal{Z}_{\ell-1} \in \mathbb{R}^{d_i \times \cdots \times d_N}$ with

$$d_i = \begin{cases} K_i & i > s_\ell \\ I_i & \text{otherwise} \end{cases} \tag{3}$$

undergoes mode-$s_\ell$ unfolding to produce matrix $\mathbf{Z}_{\ell-1}^{(s_\ell)} \in \mathbb{R}^{I_{s_\ell} \times J}$ where $J = \prod_{n \neq s_\ell} d_n$. This operation preserves cross-mode correlations while exposing the target mode's features.

2) Non-linear Projection: A dedicated multilayer perceptron processes the unfolded representation:

$$\mathbf{Z}_{\ell(s_\ell)} = \text{MLP}_{\theta_\ell}(\mathbf{Z}_{\ell-1}^{(s_\ell)}) = \text{FC}_{K_{s_\ell}}(\text{ReLU}(\text{FC}_{H_{s_\ell}}(\mathbf{Z}_{\ell-1}^{(s_\ell)}))) \tag{4}$$

where FC refers to Fully Connected layer, and the hidden dimension $H_{s_\ell}$ controls the transformation capacity. Each column vector in $\mathbf{Z}_{\ell-1}^{(s_\ell)}$ is regarded as a 'sample' during encoding.

3) Structural Reorganization: The compressed mode is folded back into tensor form $\mathcal{Z}_\ell \in \mathbb{R}^{I_1 \times \cdots K_{s_\ell} \times \cdots \times I_N}$, maintaining proper mode ordering through permutation.

The dimensionality of the tensor progressively decreases with each mode-specific mapping:

$$\mathcal{X} \xrightarrow{f_1} Z_1 \xrightarrow{f_2} Z_2 \rightarrow \cdots \xrightarrow{f_L} \mathcal{Z}_L = \mathcal{G} \tag{5}$$

After $L$ recursive stages, the process yields a compact latent core $\mathcal{G} = \mathcal{Z}_L \in \mathbb{R}^{K_1 \times \cdots \times K_N}$.

**Reverse: Pick–Unfold–Decode–Fold Recursion**. The decoder mirrors the encoding procedure in reverse order, employing distinct weights $\phi_\ell$ for each mode's reconstruction network. Correspondingly, the dimensionality of the tensor progressively increases with each mode-specific mapping:

$$\mathcal{G} \xrightarrow{g_L} \hat{\mathcal{Z}}_{L-1} \xrightarrow{g_{L-1}} \hat{\mathcal{Z}}_{L-2} \rightarrow \cdots \xrightarrow{g_1} \hat{\mathcal{X}} \tag{6}$$

This architecture generalizes Tucker decomposition by introducing learnable non-linear projections at each factorization step.

**Skip Connections for Better Optimization**. Since each mode is encoded sequentially, our method indirectly increases the network depth. To mitigate the possible vanishing gradient problem, we introduce skip connections between paired encoders and decoders. It should be noted that skip connections are only employed between hidden layers and output feature embeddings to avoid information leakage issues. The complete algorithmic workflow is presented in **Algorithm 1**.

### 3.2 Loss function and training procedure

MA-NTAE employs the same loss function as standard DAE, minimizing the reconstruction error:

$$\mathcal{L}(\theta, \phi) = \frac{1}{B} \sum_{b=1}^{B} |g_\phi(f_\theta(\mathcal{X}_b)) - \mathcal{X}_b|_F^2, \tag{7}$$

where $B$ denotes batch size. During training, the proposed model preserves the standard autoencoder training paradigm while operating directly on tensor representations.

### 3.3 Computational and Parametric Complexity

**Computational Complexity**. MA-NTAE performs *mode-wise* compression: every selected mode $s_\ell$ is first unfolded, then passes through two linear maps (*Input→Hidden→Latent*), and is finally folded back. The exact floating-point cost for this mode is

$$\text{FLOPs}_{\text{enc}}(s_\ell) = \underbrace{\mathcal{O}(I_{s_\ell} D_{-s_\ell})}_{\text{unfold}} + I_{s_\ell} H_{s_\ell} D_{-s_\ell} + H_{s_\ell} K_{s_\ell} D_{-s_\ell} + \underbrace{\mathcal{O}(K_{s_\ell} D_{-s_\ell})}_{\text{fold}} \approx D_{-s_\ell} H_{s_\ell} (I_{s_\ell} + K_{s_\ell}), \tag{8}$$

where $D_{-s_\ell} = \prod_{j \neq s_\ell} I_j$. The unfold/fold terms are linear in the element count and therefore dominated by the two matrix products in most practical settings. Summing equation 8 over all $N$ modes yields

$$\text{FLOPs}_{\text{enc}} = \sum_{s_\ell=1}^{L} H_{s_\ell} D_{-s_\ell} (I_{s_\ell} + K_{s_\ell}) = \mathcal{O}\left(L \overline{H}_{s_\ell} \overline{I}^N\right), \tag{9}$$

where $\overline{I}$ and $\overline{H}$ are the representative mode and hidden size in the regular case ($I_n = \overline{I}, H_n = \overline{H}$). The decoder is symmetric and contributes the same asymptotic cost. Therefore, in the extreme case where $L = N$, the overall complexity of MA-NTAE remains **linear in tensor order** $N$ and **proportional to each mode dimension** $I_n$.

**Parameter Complexity**. Per compressed mode $s$ the encoder holds two matrices $H_s \times I_s$ and $K_s \times H_s$ and the decoder holds their transposes, so biases aside

$$\mathrm{Params}(s) = 2H_s\big(I_s + K_s\big). \tag{10}$$

Summing over all modes gives the network size

$$\mathrm{Params_{MA\text{–}NTAE}} = 2\sum_{n=1}^{N} H_n(I_n + K_n), \tag{11}$$

linear in the tensor order $N$ and in each mode dimension $I_n$. Figure 1.1b compares the parameter growth of DAE, TFNN, and our approaches. Our method achieves substantially greater parameter efficiency compared to DAE while maintaining a marginally larger parameter count than TFNN.

## 4 EXPERIMENTS

We assess theoretical performance on synthetic tensor datasets and validate effectiveness on real-world datasets through compression experiments. We utilize PyTorch (Paszke et al., 2020) to implement our method and an NVIDIA RTX 4090 GPU to run each experiment under Windows 10 operating system. Due to the page limit, please refer to A.1 for implementation details.

### 4.1 SYNTHETIC EXPERIMENT

**Data formulation**. We synthesize $N$th-order tensors of shape $(B, I, \dots, I)$, where $B = 512$ is the batch size. The Tucker core maintains shape $B \times 0.25I \times \cdots \times 0.25I$ for consistent compression. For each sample, we generate $N - 1$ orthonormal factor matrices $\mathbf{U}^{(n)} \in \mathbb{R}^{I \times 0.25I}$ ($n = 2, \dots, N$), perturb them with Gaussian noise ($\sigma_U = 0.05$) to obtain $\tilde{\mathbf{U}}^{(n)}$, then construct clean tensors via:

$$\mathcal{X}_{\mathrm{clean}}^{(b)} = \mathcal{G}^{(b)} \times_2 \tilde{\mathbf{U}}^{(2)} \times_3 \cdots \times_N \tilde{\mathbf{U}}^{(N)}, \quad \mathcal{G}^{(b)} \sim \mathcal{N}(0,1). \tag{12}$$

We then add 30dB Gaussian noise to create $\mathcal{X}_{\mathrm{noisy}}^{(b)} = \mathcal{X}_{\mathrm{clean}}^{(b)} + \Delta$ as the input data. NMSE between $\hat{X}_{\mathrm{noisy}}$ and $X_{\mathrm{clean}}$ is computed for evaluation. For each synthetic tensor, we allocate $80\%$ of noisy samples for training and $20\%$ for testing (clean tensors split identically). For each setting of $I$, we repeat the experiment 30 times and average the results to avoid statistical bias.

**Results**. We chose DAE (Hinton & Salakhutdinov, 2006), and TFNN Chien & Bao (2018) as typical baselines representing vector-based and tensor-based neural networks, respectively. Figure 1.1c and Table 4.1 demonstrate our method's superior noise robustness and low computational cost on tensor structure recovering. The performance advantage of our method over others becomes increasingly pronounced as the dimensionality and tensor order increase. Figure 4.1 reveals that mode-shuffled samples degrade performance for all methods, with mode-wise methods (TFNN and our approach) being more sensitive to incorrect ordering. By direct non-linear tensor decomposition, our approach achieves a more stable NMSE growth trend with varying dimensionality and tensor orders while maintaining a satisfying training time.

### 4.2 EXPERIMENT ON REAL-WORLD DATA

**Compression on image datasets.** To evaluate MA-NTAE in practice, we design image compression and reconstruction experiments on multiple real-world image datasets, including three small-scale datasets (COIL-20 (Nene et al., 1996), JAFFE (Lyons et al., 1999), and Orlraws10P[1]) and two large-scale datasets CIFAR10 (Krizhevsky & Hinton, 2009) and MNIST (Deng, 2012). The information and preprocessing procedure on the datasets are detailed in Appendix A.2. We conduct standard

---

[1]https://jundongl.github.io/scikit-feature/datasets.html

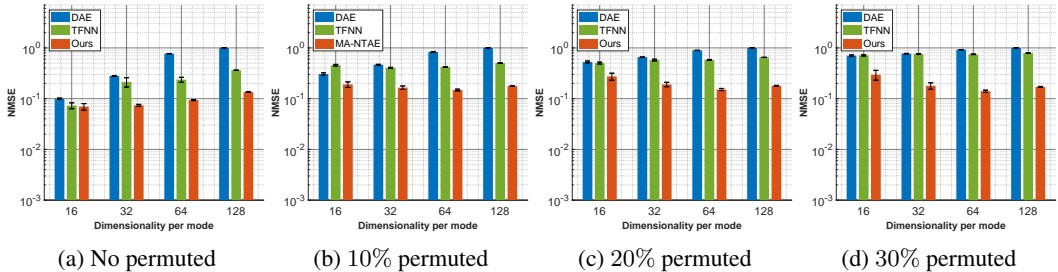

| (a) No permuted | (b) 10% permuted | (c) 20% permuted | (d) 30% permuted |

Figure 4.1: NMSE on the test set of third-order synthetic tensor data with random mode permutation. We randomly select a subset of samples, shuffle their mode orders, and evenly distribute them between the training and test sets.

Table 4.1: NMSE(±std) and training time (per epoch, seconds) on tensors of different orders. Dimension per mode is set to 20.

| Order | DAE | | TFNN | | Ours | |
|---|---|---|---|---|---|---|
| | NMSE | Time | NMSE | Time | NMSE | Time |
| 3 | $0.1467 \pm 0.0050$ | 0.0094 | $0.1249 \pm 0.0520$ | 0.0124 | $0.0743 \pm 0.0080$ | 0.0209 |
| 4 | $0.6435 \pm 0.0037$ | 0.0268 | $0.1517 \pm 0.0016$ | 0.0186 | $0.1005 \pm 0.0187$ | 0.0584 |
| 5 | $1.0023 \pm 0.0006$ | 59.2248 | $0.2870 \pm 0.0020$ | 0.4833 | $0.2440 \pm 0.0338$ | 0.5296 |

DAE (Hinton & Salakhutdinov, 2006), Convolutional AE (CAE) (Masci et al., 2011) , Variational AE (VAE) (Kingma & Welling, 2014), Tensor Factorized Neural Network (TFNN) (Chien & Bao, 2018), Tensor-train neural network (TTNN) (Novikov et al., 2015), and T-SVDM neural network (TMNN) Newman et al. (2024) for comparison. According to the experimental results, our method achieves: 1) superior dimensionality reduction and recovery (Figure 4.2) with lower reconstruction error (Table 4.2 and Figure 4.3b), 2) more stable training convergence with better generalization (Figure 4.3c and 4.3d), and 3) relatively less training time and parameters (See Table 4.3). Overall, our method achieves a favorable balance between reconstruction performance and computational complexity.

**Ablation Study.** An ablation study is carried out to evaluate the impact of skip connections and mode order on model performance. MA-NTAE is trained under different skip connection settings and mode encoding orders, with NMSE computed on the test sets. Results in Table 4.4 show that skip connections enhance autoencoding performance, yielding lower and more stable reconstruction errors. Due to the non-commutative nature of mode-wise encoding, different mode orders affect performance, implying that the processing sequence should be tailored to specific applications. Figures 4.4a, obtained by repeated experiments on JAFFE, indicate that skip connections prevent severe performance degradation despite reduced feature space capacity, suggesting they stabilize the optimization process rather than simply bypassing information. Even without skip connections, MA-NTAE can still achieve great generalization on the test set (See Figure 4.4b retrieved at CR $\approx$ 39), which indicates our method's outstanding tensorial learning ability.

Table 4.2: Best reconstruction NMSE in the compression experiment corresponding to Figure 4.2. The bold and underlined entries highlight the smallest and second-smallest values, respectively.

| Dataset | Ours | DAE | TMNN | TFNN | TTNN | CAE | VAE |
|---|---|---|---|---|---|---|---|
| COIL20 | **0.0039** | 0.0581 | 0.0507 | 0.0138 | 0.0173 | **0.0039** | 0.0374 |
| JAFFE | **0.0046** | 0.0318 | 0.0268 | 0.0126 | 0.0186 | 0.0069 | 0.0319 |
| Orlraws10P | **0.0042** | 0.0450 | 0.0426 | 0.0143 | 0.0237 | 0.0054 | 0.0519 |
| MNIST | **0.0098** | 0.0268 | 0.1248 | 0.0675 | 0.0697 | 0.0204 | 0.0593 |
| CIFAR10 | **0.0293** | 0.0620 | 0.2462 | 0.1190 | 0.0647 | 0.0564 | 0.0908 |

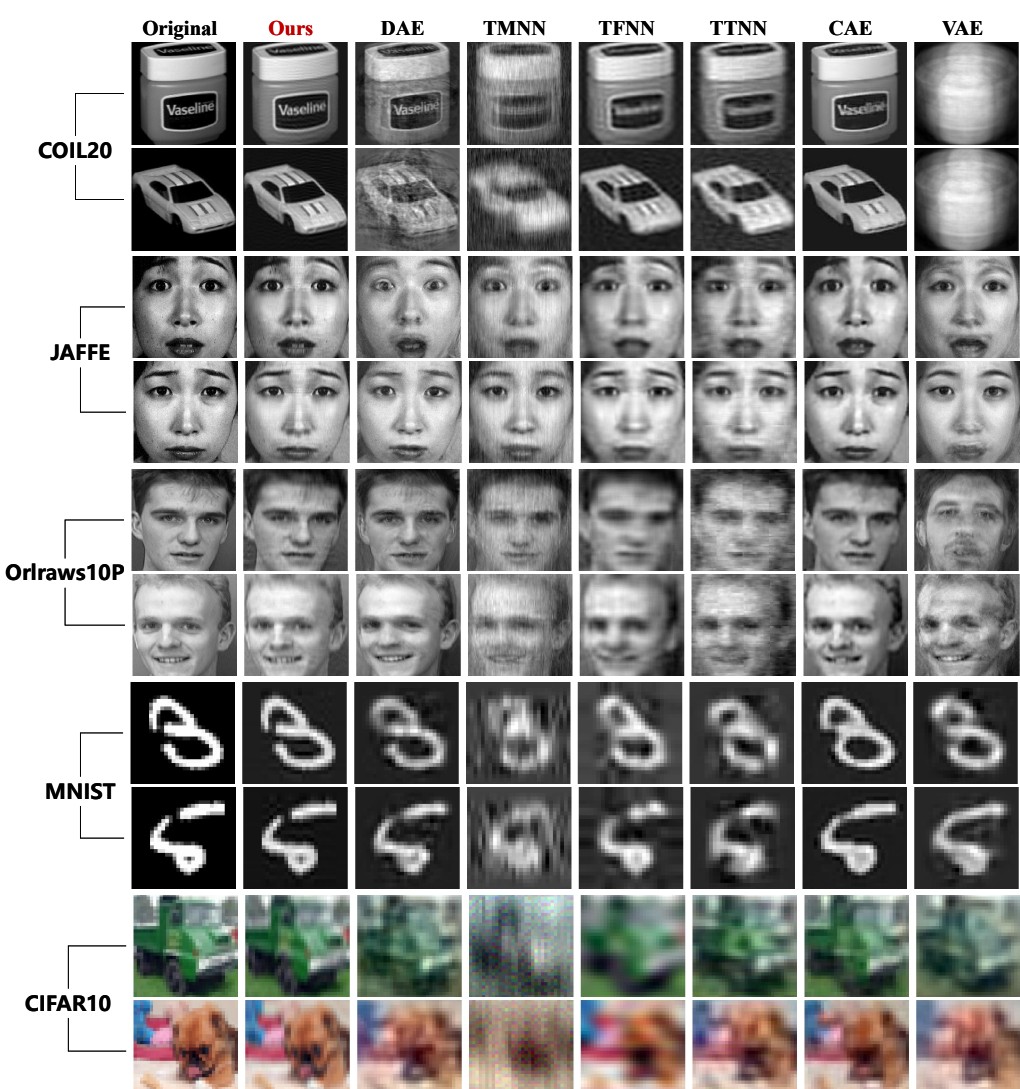

Figure 4.2: Reconstruction results on the test sets of real-world datasets. Two-layer dimensionality reduction (See Appendix A.1) with factors of $0.5$ for each mode per layer is applied for all models on each dataset. Our method combines the ability of TNNs to maintain the image structure (camera viewpoint and target orientation) with the capability of MLPs to recover fine details. See Appendix A.3 for more results.

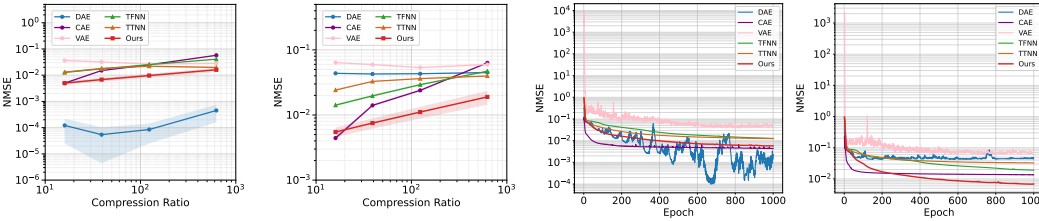

(a) CR vs. Training NMSE    (b) CR vs. Testing NMSE    (c) Training NMSE curve    (d) Testing NMSE curve

Figure 4.3: Results on Orlraws10P with different Compression ratios (CR). The CRs are varied by setting the dimensionality reduction factor in the range of $[0.5, 0.4, 0.3, 0.2]$. The experiment under each CR is repeated for 30 times. (a) and (b) record the best NMSE during training and testing, respectively. (c) and (d) are retrieved at the training process when CR $\approx 39$.

Table 4.3: Cost of training time and parameters in the compression experiment. For each dataset, the first and second rows provide the training time (seconds) per epoch and the parameter numbers, respectively. The bold and underlined entries highlight the smallest and second-smallest values, respectively.

| Dataset | Ours | DAE | TMNN | TFNN | TTNN | CAE | VAE |
|---|---|---|---|---|---|---|---|
| COIL20 | 0.0339 | 0.6809 | 0.0398 | **0.0308** | 0.1598 | 0.1217 | 0.7046 |
| | 41537 | 142631936 | 1139712 | 20480 | 34368 | **2633** | 146843648 |
| JAFFE | 0.0087 | 0.2231 | 0.0110 | **0.0082** | 0.0299 | 0.0171 | 0.2308 |
| | 41537 | 142631936 | 1139712 | 20480 | 34368 | **2633** | 146843936 |
| Orlraws10P | 0.0077 | 0.0893 | 0.0090 | 0.0065 | 0.0167 | **0.0060** | 0.0945 |
| | 26720 | 56420196 | 629188 | 13130 | 39960 | **2633** | 58090088 |
| MNIST | 0.9501 | **0.5635** | 1.2148 | 1.0757 | 1.7186 | 0.6916 | 0.8926 |
| | 2087 | 327761 | 12572 | **982** | 3394 | 2633 | 338198 |
| CIFAR10 | 1.1534 | 0.9629 | 1.0938 | 0.9992 | 1.8617 | **0.7258** | 1.2476 |
| | 2753 | 5018304 | 57024 | **1298** | 15568 | 3211 | 5169024 |

Table 4.4: Ablation results on JAFFE, Orlraws10P, and COIL20 (NMSE($\pm$std)).

| Skip Connection | Mode Encoding Order | Dataset | | |
|---|---|---|---|---|
| | | JAFFE | Orlraws10P | COIL20 |
| w | $[2,3]$ | $0.0048 \pm 0.0003$ | $0.0051 \pm 0.0012$ | $0.0050 \pm 0.0006$ |
| | $[3,2]$ | $0.0055 \pm 0.0004$ | $0.0068 \pm 0.0007$ | $0.0021 \pm 0.0001$ |
| w/o | $[2,3]$ | $0.0143 \pm 0.0007$ | $0.0192 \pm 0.0008$ | $0.0165 \pm 0.0017$ |
| | $[3,2]$ | $0.0144 \pm 0.0009$ | $0.0187 \pm 0.0018$ | $0.0143 \pm 0.0009$ |

# 5 CONCLUSION

In this work, we address the challenges of deep learning on high-order tensor data by proposing the Mode-Aware non-linear Tucker Autoencoder (MA-NTAE), a novel framework that integrates classical Tucker decomposition with modern autoencoding techniques through recursive Pick-Unfold-Encode-Fold operations and enables flexible mode-aware processing of tensor data. Compared to DAE variants and existing tensorial networks, our approach achieves superior reconstruction accuracy with relatively small parameter sizes and training time across simulated and real-world tensor data of varying orders and dimensions. The ablation study shows the importance of the introduced skip connection and MA-NTAE's sensitivity to mode orders.

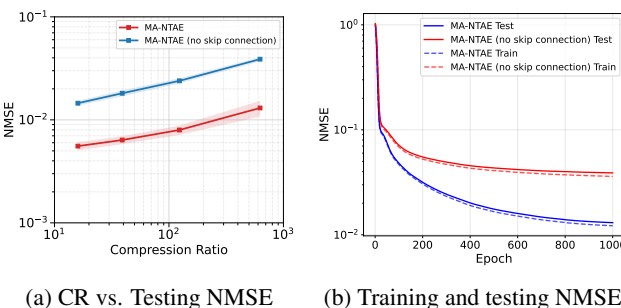

(a) CR vs. Testing NMSE  (b) Training and testing NMSE

Figure 4.4: Comparison of NMSE curves for MA-NTAE with and without skip connections.

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

# A  APPENDIX

## A.1  IMPLEMENTATION DETAILS OF COMPARATIVE METHODS

The implementation details of comparative methods are listed as follows:

- **Unified Structure**: All methods employ an identical two-layer encoder-decoder architecture, as illustrated in Figure A.1. The input data dimensions are specifically tailored to each model's architecture. We consistently employ the Rectified Linear Unit (ReLU) function as the activation function for all methods. In the experiments, we consistently employ an end-to-end training strategy (without using stacked training) to ensure fairness, while also demonstrating the training-friendly advantage of our approach. The learning rate is fixed to $10^{-3}$. All models are trained for 1000 epochs. And the batch size is set according to the sample size of each dataset in the experiments (See Table A.3).

- **Loss function.** All models optimize the Mean Squared Error (MSE) as the loss function, while the normalized MSE (NMSE) is utilized for evaluation. Specifically, for the VAE, we employ a combination of MSE and KL divergence as the loss function to align with its underlying principle.

- **MA-NTAE.** The inputs and outputs of our method strictly adhere to the structure defined in Figure A.1 across most benchmark datasets, encoded following the modal order of $[2, 3]$.

However, for the CIFAR10 dataset, where the input format is Batch × Channel × Height × Width—we make an exception by preserving the dimensionality of the channel mode. In this case, the reduction is applied to the modes in the order of $[3, 4, 2]$, corresponding to [Height, Width, Channel].

- **DAE and VAE.** The input dimensions for the corresponding layers in DAE and VAE are structured as $B \times \prod_i I_i - B \times \prod_i \alpha I_i - B \times \prod_i \alpha^2 I_i - B \times \prod_i \alpha I_i - B \times \prod_i I_i$.

- **CAE.** For CAE, we utilize a $3 \times 3$ convolutional kernel with a stride of $1$ and padding of $1$. Dimensionality reduction in the encoder is achieved through max pooling layers, while upsampling operations are employed in the decoder for dimensionality expansion. The first encoder layer outputs $16$ channels, and the second encoder layer produces $8$ channels. The decoder follows a symmetric channel configuration relative to the encoder. For grayscale image inputs, we unsqueeze an additional channel dimension. In experiments involving compression ratio variations, we replace the pooling layers with adaptive pooling to align the compression ratios across all methods, thereby enabling more flexible control over the capacity of the output feature space.

- **TFNN.** The TFNN employs the same input and output configuration for each layer as our method.

- **TTNN.** Since the TTNN operates by applying a tensor-train (TT) decomposition to the weight matrices of the linear layers in the DAE, the TT-format is configured individually for different input data sizes. The specific settings of each encoder layer are detailed in Table A.1. For different compression ratios, the corresponding settings are given in Table A.2. The decoder layers have a symmetrical architecture to the encoder layers. Please refer to Novikov et al. (2015) for technical details.

Table A.1: The specific settings of each encoder layer in TTNN on different datasets, including the input size, output size, and weight matrices' TT-ranks in each layer. Each upper row and bottom row within a dataset represent the size of the reshaped input and output data, respectively. The dimensionality reduction factor $\alpha$ for each mode in the original data is set to $0.5$.

| Dataset | First layer | Second layer | TT-ranks |
|---------|-------------|--------------|----------|
| COIL20 | $8 \times 8 \times 4 \times 4 \times 4 \times 2 \times 2$ | $8 \times 4 \times 4 \times 4 \times 2 \times 2 \times 2$ | $1 \times 8 \times 8 \times 8 \times 8 \times 8 \times 8 \times 1$ |
| | $8 \times 4 \times 4 \times 4 \times 2 \times 2 \times 2$ | $4 \times 4 \times 4 \times 2 \times 2 \times 2 \times 2$ | $1 \times 8 \times 8 \times 8 \times 8 \times 8 \times 8 \times 1$ |
| JAFFE | $8 \times 8 \times 4 \times 4 \times 4 \times 2 \times 2$ | $8 \times 4 \times 4 \times 4 \times 2 \times 2 \times 2$ | $1 \times 8 \times 8 \times 8 \times 8 \times 8 \times 8 \times 1$ |
| | $8 \times 4 \times 4 \times 4 \times 2 \times 2 \times 2$ | $4 \times 4 \times 4 \times 2 \times 2 \times 2 \times 2$ | $1 \times 8 \times 8 \times 8 \times 8 \times 8 \times 8 \times 1$ |
| Orlraws10P | $8 \times 8 \times 7 \times 23$ | $8 \times 4 \times 7 \times 23$ | $1 \times 8 \times 8 \times 8 \times 1$ |
| | $8 \times 4 \times 7 \times 23$ | $4 \times 2 \times 7 \times 23$ | $1 \times 8 \times 8 \times 8 \times 1$ |
| MNIST | $14 \times 14 \times 4$ | $7 \times 7 \times 4$ | $1 \times 4 \times 4 \times 1$ |
| | $7 \times 7 \times 4$ | $3 \times 3 \times 4$ | $1 \times 2 \times 2 \times 1$ |
| CIFAR10 | $16 \times 16 \times 3 \times 4$ | $8 \times 8 \times 3 \times 4$ | $1 \times 8 \times 8 \times 8 \times 1$ |
| | $8 \times 8 \times 3 \times 4$ | $4 \times 4 \times 3 \times 4$ | $1 \times 4 \times 4 \times 4 \times 1$ |

- **TMNN.** Due to its foundation on the T-SVDM principle for feature encoding, TMNN can only reduce the dimensionality along one mode of the three-dimensional input. To ensure alignment of its feature space with other methods, we specifically adjust the dimensionality reduction factor for TMNN. For instance, on the COIL20 dataset, the output after the first encoder is typically $\alpha 128 \times \alpha 128$ for other methods, whereas for TMNN it becomes $\alpha^2 128 \times 128$. The same adjustment logic applies to other datasets. We set the minimum mode reduced dimensionality to $1$. This constraint is reasonable because the dimensionality of the other mode remains unchanged during encoding. However, due to the difficulty in precisely controlling the compression ratio of TMNN, it was excluded from the experiments involving compression ratio variations. This was done to prevent unfair comparisons resulting from discrepancies in feature space capacity.

Table A.2: The specific settings of each encoder layer in TTNN on Orlraws10P with different dimensionality reduction factors $\alpha$.

| $\alpha$ | First layer | Second layer | TT-ranks |
|---|---|---|---|
| 0.5 | $8 \times 8 \times 7 \times 23$ | $8 \times 4 \times 7 \times 23$ | $1 \times 8 \times 8 \times 8 \times 1$ |
|  | $8 \times 4 \times 7 \times 23$ | $4 \times 2 \times 7 \times 23$ | $1 \times 8 \times 8 \times 8 \times 1$ |
| 0.4 | $64 \times 7 \times 23$ | $9 \times 11 \times 16$ | $1 \times 8 \times 8 \times 1$ |
|  | $9 \times 11 \times 16$ | $2 \times 7 \times 17$ | $1 \times 8 \times 8 \times 1$ |
| 0.3 | $64 \times 7 \times 23$ | $9 \times 9 \times 11$ | $1 \times 8 \times 8 \times 1$ |
|  | $9 \times 9 \times 11$ | $9 \times 4 \times 2$ | $1 \times 8 \times 8 \times 1$ |
| 0.2 | $112 \times 92$ | $18 \times 22$ | $1 \times 8 \times 1$ |
|  | $18 \times 22$ | $4 \times 3$ | $1 \times 8 \times 1$ |

## A.2 DATASETS INTRODUCTION

Table A.3 details the characteristics of the four real-world image datasets used in our experiments, including three small-scale and two large-scale datasets. The descriptions are given as follows:

- **COIL20** (Nene et al., 1996), comprises grayscale images of 20 distinct objects, captured through rotational photography at 5-degree intervals (yielding 72 images per object). All images are standardized to $128 \times 128$ pixels. This dataset consists of two subsets: The first subset contains 720 raw images covering 10 object categories, while the second subset includes $1,440$ preprocessed images encompassing all 20 object categories.

- **JAFFE** (Lyons et al., 1999), comprises 213 facial expression images captured in a laboratory setting. All images are standardized to $128 \times 128$ pixels. Ten Japanese female participants were instructed to perform seven prototypical expressions (six basic emotions plus neutral), with high-resolution photographs taken of each expression.

- **Orlraws10P**[2], contains 100 standardized grayscale images ($112 \times 92$ pixels) derived from the ORL face database (Samaria et al., 1994), sampling 10 randomly selected subjects. Each subject's 10-image sequence captures controlled intra-subject variations, including: temporal acquisition differences, facial expressions (eye openness and smile states), lighting conditions, and accessory modifications (glasses on/off).

- **MNIST** (Deng, 2012) The MNIST dataset is a foundational benchmark in computer vision. It consists of 70,000 grayscale images of handwritten digits (0-9), split into 60,000 training and 10,000 test images. The digits are centered within each $28 \times 28$ pixel image, showcasing a variety of handwriting styles in a clean, pre-processed format.

- **CIFAR10** (Krizhevsky & Hinton, 2009) The CIFAR10 dataset comprises 60,000 $32 \times 32$ color images in 10 classes, such as airplanes, cars, animals (like birds, cats, deer, dogs, frogs, horses), and vehicles (ships, trucks). It is partitioned into a standard training set of 50,000 images and a test set of 10,000 images.

We randomly used $50\%$ of the samples as the training set and the remaining $50\%$ as the test set, except for MNIST and CIFAR10, which have officially designated training and test sets. The preprocessing for all datasets only involve normalization, along with the necessary shuffling of sample order during the training phase.

## A.3 SUPPLEMENTARY EXPERIMENTAL RESULTS

The supplementary results for the compression experiment are given in Figure A.2, Figure A.3, Figure A.4, Figure A.5, and Figure A.6. Overall, vector-based methods (DAE and VAE) tend to learn "averaged" encodings, which often leads to confusion between targets or viewpoints on the test set, especially on small-scale datasets like Orlraws10P and COIL20. Conversely, methods that account

---

[2]https://jundongl.github.io/scikit-feature/datasets.html

Table A.3: Dataset Statistics

| Dataset | #Sample | #Feature | #Class | Batch size |
|---------|---------|----------|--------|------------|
| COIL20 | 1440 | $128 \times 128$ | 20 | 64 |
| JAFFE | 213 | $128 \times 128$ | 7 | 32 |
| Orlraws10P | 100 | $92 \times 112$ | 10 | 16 |
| MNIST | 70000 | $28 \times 28$ | 10 | 128 |
| CIFAR10 | 60000 | $32 \times 32 \times 3$ | 100 | 128 |

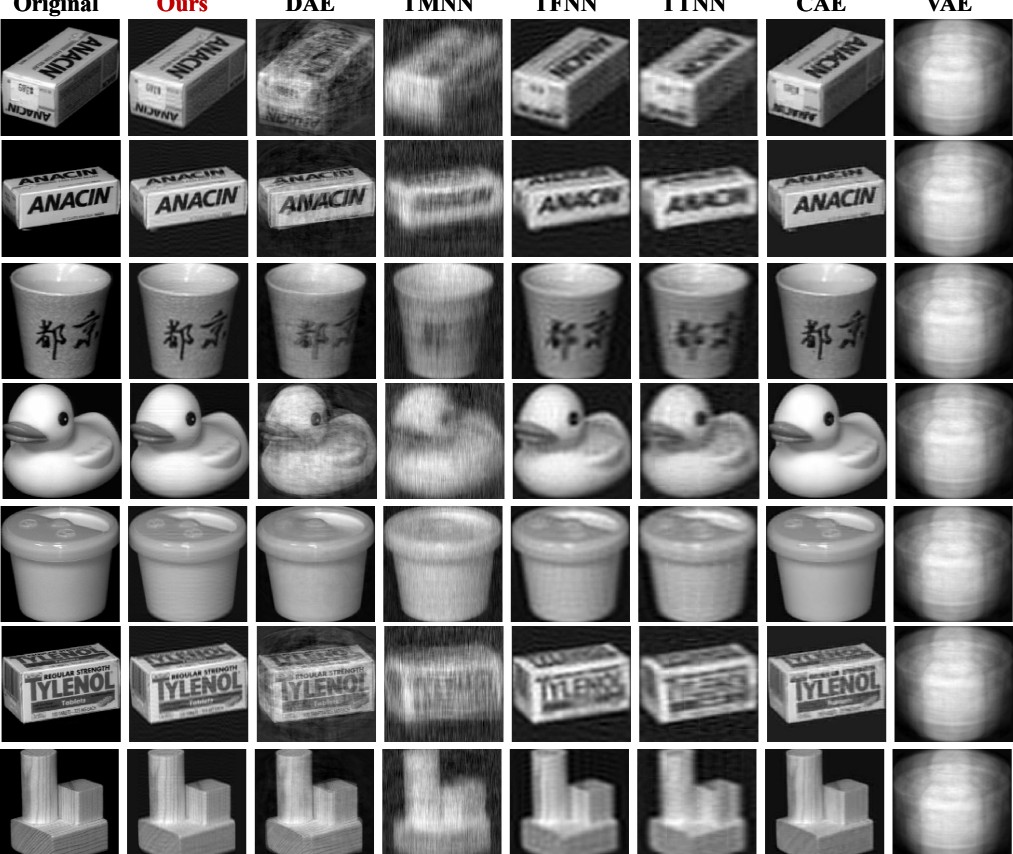

Figure A.2: Reconstruction results on the test set of COIL20.

for tensor structure (TMAE, TFAE, TTAE, CAE, and our approach) demonstrate better generalization. Furthermore, thanks to our Pick-Unfold-Encode-Fold architecture's enhanced capability in modeling non-linear relationships, it achieves superior reconstruction of image details.

## A.4 LARGE LANGUAGE MODEL USAGE DECLARATION

Our use of Large Language Models (LLMs) was limited to polishing sentences and fixing grammar, with no impact on the research content.

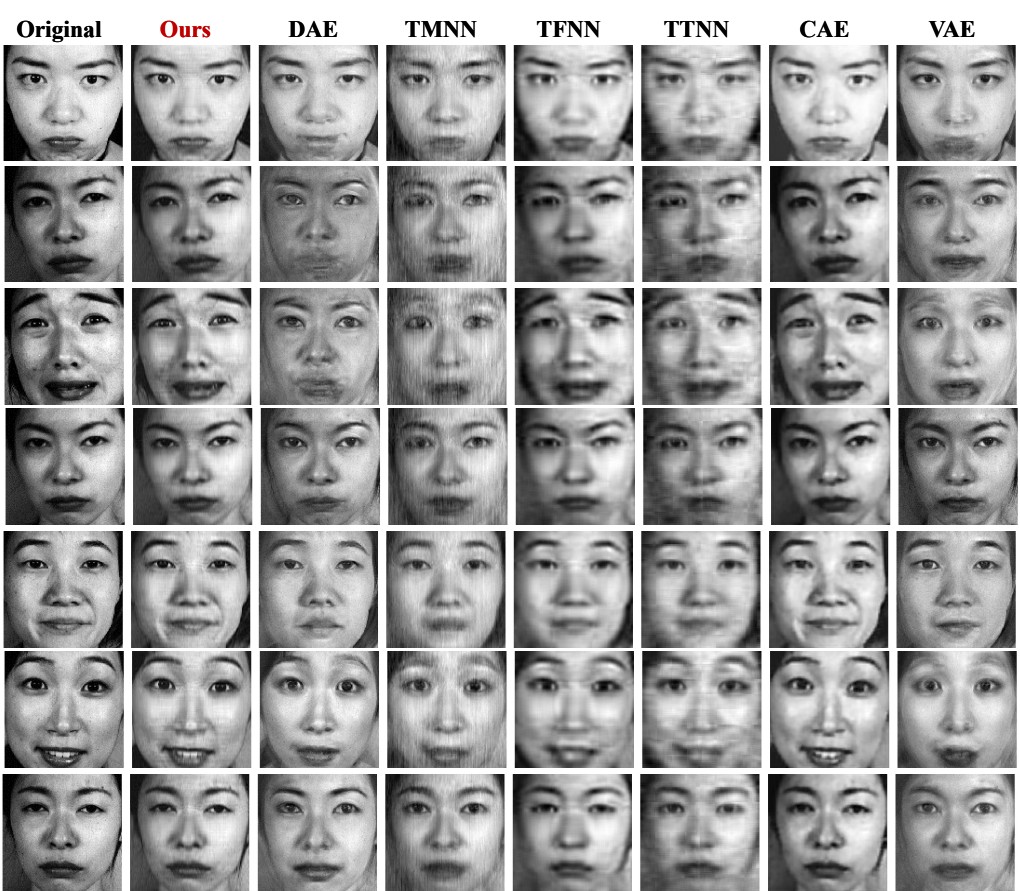

Figure A.3: Reconstruction results on the test set of JAFFE.

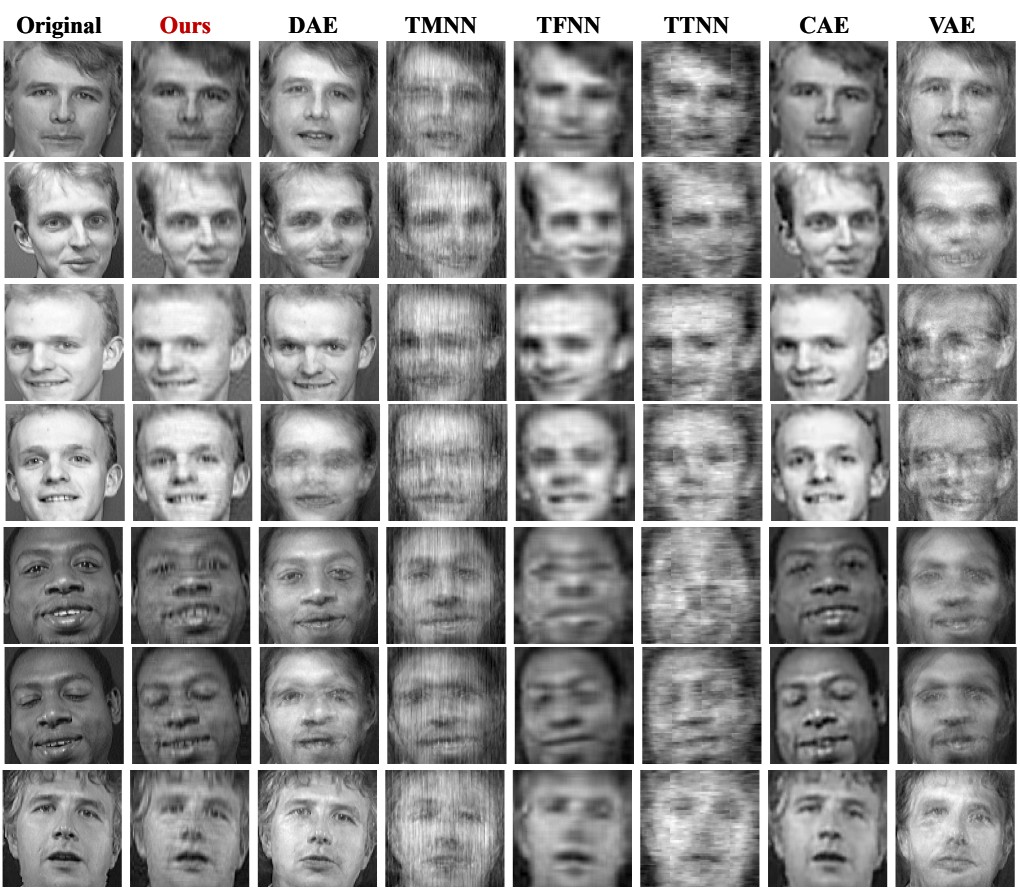

| Original | Ours | DAE | TMNN | TFNN | TTNN | CAE | VAE |

Figure A.4: Reconstruction results on the test set of Orlraws10P.

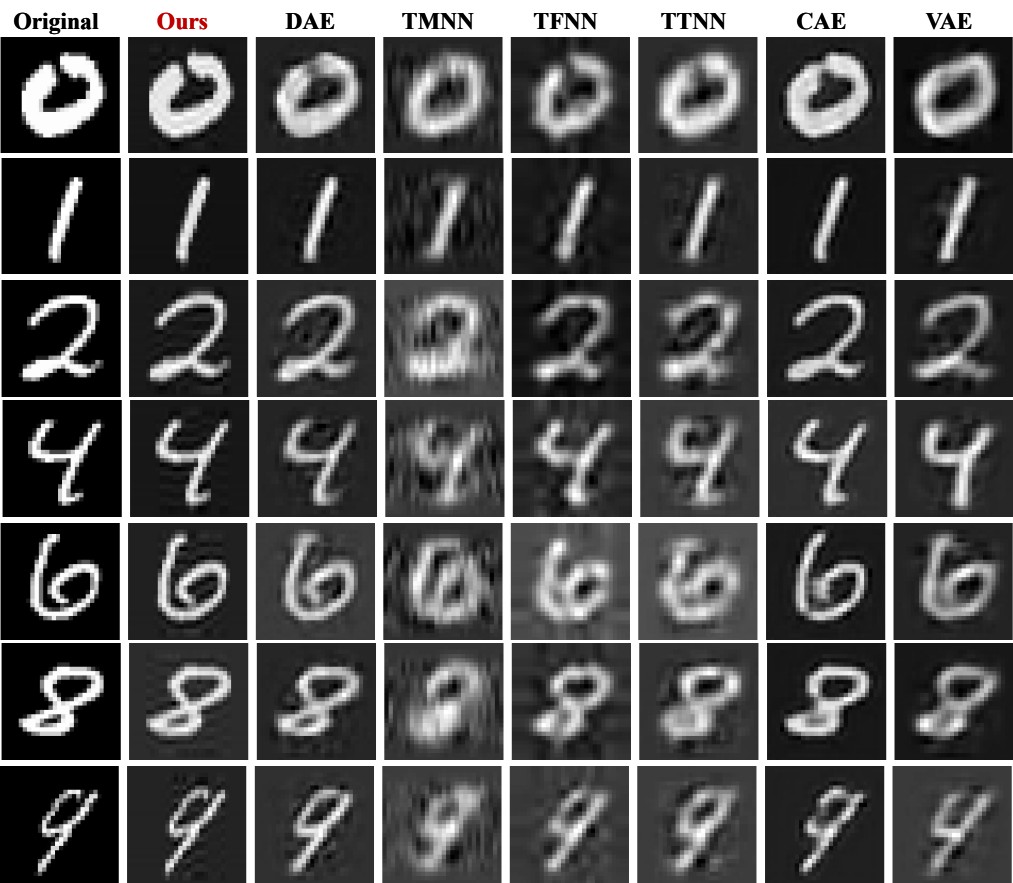

Figure A.5: Reconstruction results on the test set of MNIST.

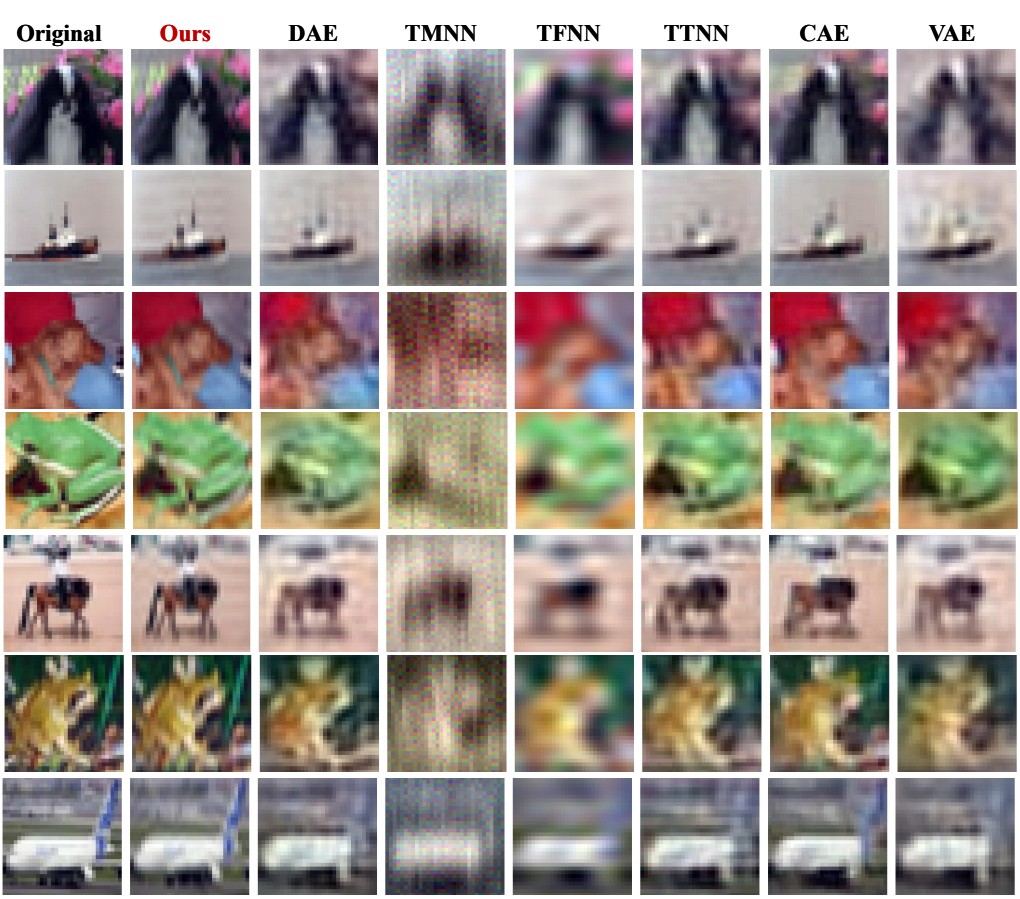

Figure A.6: Reconstruction results on the test set of CIFAR10.

