# OpenReview forum: "Pick‑and‑Unfold: Mode‑Aware Non‑Linear Tucker Autoencoder for Tensorial Deep Learning"
_ICLR.cc/2026/Conference — Submitted to ICLR 2026_

### Official Review · Reviewer_M8ZQ · 2025-10-17

**Soundness:** 2
**Presentation:** 2
**Contribution:** 1
**Rating:** 2
**Confidence:** 4

**Summary:**

This work studies an AE structure for tensor data. The authors unfold input tensors along each mode, and apply MLP on the unfolded data. These MLPs are applied on each mode in sequence, and result in a smaller tensor latent variable, which resembles traditional AE. Then, this tensor latent variable is decoded into the data space in a reverse manner. The model is trained by MSE. The authors compare the proposed model with several AE and Tensorized AE baselines on several image datasets.

**Strengths:**

The overall structure is clear.

**Weaknesses:**

**Methodology:** The novelty and contribution are limited. There are many works considering the AE-like structure for tensor data. Just to name a few, the mode-specific MLP in this work is similar to [1-4]. There are certainly differences. But I think the differences are not fundamental. So the literature review is also inadequate.

- [1]. Deng, Z., et al. Factorized variational autoencoders for modeling audience reactions to movies. In CVPR 2017.
- [2]. Fan, J. Multi-mode deep matrix and tensor factorization. In ICLR 2022.
- [3]. Tao, Z., et al. Nonparametric tensor ring decomposition with scalable amortized inference. Neural Networks 2024.
- [4]. Iwata, T., & Kumagai, A. Meta-learning from Heterogeneous Tensors for Few-shot Tensor Completion. In AISTATS 2025.

**Experiments:** All datasets are small. The authors say `two large-scale datasets CIFAR10 and MNIST`. It is hard to consider these datasets as _large_ in 2025. The baselines are not competitive and I highly suspect that the baselines are not optimized well. For example, the CAE only has two convolutional layers and max pooling layers. Apparently we can train much better AEs and VAEs on these datasets nowadays. Furthermore, as I mentioned before, there are many AEs for tensor data that can be baselines. Finally, regarding the generalization in Figure 4.3, currently NNs rely heavily on compound techniques of data augmentations, optimization, and regularizations. I do not think the comparison here is promising.

**Minor:**

- Notation inconsistency. In Eq 5, should $L$ equal to $N$?
- In Line 241, I do not find Algorithm 1.

**Questions:**

See weaknesses

---

### Official Review · Reviewer_4iKw · 2025-10-26

**Soundness:** 2
**Presentation:** 1
**Contribution:** 2
**Rating:** 2
**Confidence:** 4

**Summary:**

This paper introduced a nonlinear Tucker-based AE by sequentially unfold-encode-fold procedure. The authors claimed that by adopting the unfolding and encode procedure, it can preserve and maintain mode-interactions between different modes. Experiments on several image datasets and synthetic data demonstrates the effectiveness of the proposed method.

**Strengths:**

- Strengths

This paper introduced a strategy for a tensor-based encoder, which first unfolds the data on one mode, and then encodes the data on the specific mode. This is quite interesting since it is quite different from the vanilla AE by simply unfolding the data into a vector.

**Weaknesses:**

- Weakness

1. The novelty of this paper seems limited, as it only conducts the fold and unfolding operator before and after the MLP layer. The advantages or the significance of these two operators are also not clear, and have not been clearly demonstrated in the paper.

2. The authors claimed that TFNN fails to model the non-linear interactions between different modes. However, I think TFNN can do that since it have individual mode-wise mapping using U and cross-mode interaction using G. Therefore, it will be better if the authors can present more evidence or clarity on this claim.

3. About the motivations and significance of this paper. I don’t think this paper presents a very clear presentation. Compared to the vanilla DAE, I think the main difference or contribution are on the folding and unfolding step after the MLP layer. This folding and unfolding can be seen as a non-parametric mapping from one matrix to another matrix. Maybe the authors can re-consider to re-build the motivation and whole framework for this paper.

4. In the experiment section, the datasets are too old or say to standard. It will be suitable for linear model such as Tucker decomposition. However, for nonlinear DAE, I think it would be better to evaluate its performance on more popular datasets used in AE such as cifa100 and ImageNet (or the subset of ImageNet), it would be a better choice to demonstrate the expressiveness of the proposed DAE.

**Questions:**

See the weakness section for details.

---

### Official Review · Reviewer_QV6s · 2025-11-01

**Soundness:** 2
**Presentation:** 2
**Contribution:** 1
**Rating:** 2
**Confidence:** 2

**Summary:**

The paper proposes MA-NTAE model: using neural network to do tucker decomposition like autoencoder. For input high order tensors, it repeats pick-unfold-encode-fold step for each mode to encode, and then reverses it to decode. The proposed model keeps the parameter/compute grow linearly w.r.t. tensor orders and mode sizes. The authors evaluated MA-NTAE on image reconstruction tasks and compared with auto encoder models and three tensor neural network models (TFNN, TTNN, TMNN).

**Strengths:**

* The pick–unfold–encode–fold recursion is a neat technique for applying tucker like tensor decomposition with neural nets
* The model's linear growth on the number of parameters/compute is a nice-to-have-feature

**Weaknesses:**

* Novelty of the proposed model is not well demonstrated. Are there anything that the proposed model can learn but the existing tensor neural network models cannot? If so, why?
* Evaluation scope is very limited. All real world data are just order-3 images and no higher order data is being tested. Only reconstruction objectives are tested. It would be good to have more diversed evaluation setups to convince readers the full effectiveness of the proposed models.
* The experiment results implied that performance depends on the processing order. There's not too much discussions on how to handle this practically.

**Questions:**

See weakness parts.

---

### Official Review · Reviewer_STxn · 2025-11-06

**Soundness:** 2
**Presentation:** 1
**Contribution:** 2
**Rating:** 2
**Confidence:** 4

**Summary:**

This paper uses encoders to replace the mode-n product in Tucker decomposition mode-by-mode. Experiments on synthetic data and vision data show parameter efficiency (and thus running time) in comparision with many non-linear variants of tensor decompositions.

**Strengths:**

1. A new non-linear extension of the Tucker decomposition is proposed within the contex of tensor-factorized neural networks.
2. Experimental results (exampled by Figure 1.1) illustrate the efficiency of this non-linear Tucker variant.

**Weaknesses:**

Weakness 1. Although I think the proposed model is new, the idea behind it — using encoders/decoders to replace the linear transform — is far from novel. Specifically, replacing mode-n product by non-linear transform is not new. For example, similar ideas can be found in [1].

[1] Varolgunes, Uras, et al. NMTucker: Non-linear Matryoshka Tucker Decomposition for Financial Time Series Imputation. ICAIF 2023.

Weakness 2. Introducing mode-by-mode non-linear transform to a tensor may also introduce the non-commutative issue: the result of "first mode-2 then mode-3" may differ from the result of "first mode-3 then mode-2". I am not quite sure whether this is exactly a serious problem in this work, but I think this issue deserves a discussion.

Weakness 3. The experimental results needs further varification. The most significant contribution of this work is its parameter efficiency in comparison with other neural networks based on non-linear tensor decompositions. However, the experiments involve so many details, and it is not easy to reproduce the experimental results without a very comprehensive implementation shared from the authors. Moreover, some results seem too good (eg. see Table 4.2), which I am not sure whether the comparison is sufficiently fair.

Weakness 4. The writting is not always satisfactory. For example, in Eq. (7), the norm symbol should be $\\|\cdot\\|$ instead of $|\cdot|$. This seems a common error when copying equations from LLM's output.

**Questions:**

1. Regarding Weakness 1, what is the most significant novelty of the proposed model?

2. Can you discuss the non-communatitive issue in Weakness 2?

3. In Figure 4.1, a more clear explanation of the four subplots is needed. The captions are too concise.

---

### Meta-Review · Area_Chair_w4ZQ · 2026-01-09

**Summary:**

All reviewers recommend rejection (2), and not author rebuttal has been posted.
Main concerns: not a novel idea to replace Tucker decomposition by a learned autoencoder and experiments are weak.

**Reviewer Concerns:**

No rebuttal.

**Reviewer Scores:**

2, 2, 2, since there is no rebuttal - no change at all.

---

### Decision · Program_Chairs · 2026-01-26

Reject